# Study on the Influence Mechanism of Intangible Cultural Heritage Distribution from Man–Land Relationship Perspective: A Case Study in Shandong Province

**Lin Meng** [1], **Chuanguang Zhu** [2], **Jie Pu** [3], **Bo Wen** [3] and **Wentao Si** [1,*]

1    School of Public Administration, Shandong Normal University, Jinan 250014, China; menglin@sdnu.edu.cn
2    Jinan Institute of Surveying and Mapping, Jinan 250100, China; 2019109072@stu.njau.edu.cn
3    College of Landscape Architecture, Nanjing Forestry University, Nanjing 210037, China;
      ppuuyyuu@njfu.edu.cn (J.P.); wenbo2019@njfu.edu.cn (B.W.)
*    Correspondence: siwentao@sdnu.edu.cn

**Abstract:** Spatial autocorrelation, cold and hot spot analysis, and standard deviation ellipse analysis were used to analyze the spatial distribution of intangible cultural heritage (ICH). Geodetectors were used to reveal the factors that influenced the distribution in Shandong Province. The results showed that: (1) The ICH in Shandong Province covered most ICH types with the difference in the number of expressions of ICH of a different type. Traditional artistry, traditional art, traditional sports, recreation and acrobatics, and folk literature are the main types of ICH. (2) The spatial distribution of ICH showed a great difference. Multiple concentration areas and deficient areas were presented that followed the direction from southwest to northeast. (3) Man–land relationship-related factors such as population, waters, urban–rural size, and air temperature showed important influence on ICH distribution. The influence of interaction among influence factors is higher than a single factor. In summary, man–land relationships are the key factors that influenced ICH distribution.

**Keywords:** intangible cultural heritage; man–land relationship; influence mechanism; Shandong Province; China

## 1. Introduction

ICH is the traditional cultural expressions and cultural spaces which are closely related to the life of people of all generations. ICH recorded the process by which humans altered nature, carried human history and civilization, and condensed the essence of traditional culture [1]. Meanwhile, elements of ICH which show distinct national characteristics are important vectors of the humanistic spirit for a nation, as well as the symbol of cultural identity [2]. However, with the rapid development of modern society, ICH is under the threat of loss, destruction, and drainage away to foreign countries. What is more, ICH abuse and exploitation frequently occur [3]. Due to the limitations of ICH application and judgment, some expressions of ICH were neglected because they could not bring economic income in a short time. The lack of investment and inheritance made things even worse. Some expressions of ICH may gradually lose their existing conditions and disappear [4]. The ICH distribution and their influence factors were of great concern recently because they were easily influenced by external factors.

The current study begins with multisource data, applying methods such as spatial autocorrelation, hot spot clustering, location entropy, and kernel density estimation to quantitatively analyze the spatial distribution of ICH on a different scale. At the national scale, Chinese ICH showed clear band shapes and group distribution. The ICH showed compact distribution in the east and south of China while a sparse distribution pattern was shown in the north and west of China. Among the numerous expressions of ICH, traditional drama and traditional artistry are the main types [5]. On the provincial scale,

ICH in Guizhou Province [6], Shanxi Province [7], and other provinces showed grouped patterns of band shape and foliate distribution. The main ICH type varied among different provinces. On a city scale, the distribution of the intangible cultural heritage of music in Xiangxi had spatial heterogeneity, and agglomeration appeared in some areas [8].

In the influence mechanism study, the distribution of ICH was influenced by the geographic environment, as well as human life and production, closely related to social, economic, and natural factors. Factors such as economic development level [9,10], urbanization [11], ethnic minorities [12], transportation [13], population size [14], topography [15], and water [16] posed an important influence on the spatial distribution of ICH. Previous studies reported that the interactions among factors have a potential influence on the distribution of ICH as well [17]. What is more, possibilism proposed that the natural environment provides various possibilities for human activity, and humans themselves and their selection made such kinds of possibilities turn into reality [18]. Adaptation theory demonstrated that through the development of culture, humans could adapt to the natural environment [18]. Cultural determinism points out that the natural environment greatly affected human activity at the early stage of human history or when cultural development speed was slow. The development of productivity enhanced the ability of a human to control the environment [18]. Therefore, ICH is the external expression of coordinated development of the man–land relationship. ICH is one kind of continuously improved culture that superimposed the natural environment at different historical stages and fulfilled historical accumulation through humans [19]. It has been indicated that the ICH distribution and influence factors have significant differences among different regions. However, the previous research only highlighted the identification of ICH agglomeration areas while ignoring the lack of regional identification. The influence mechanism analysis is mainly based on qualitative methods and less on quantitative methods. Additionally, influence indicators are simply selected from the society–economy–nature perspective, without considering the man–land relationship.

Shandong Province is located in east China, with the Yellow River flowing across the whole province from west to east. Throughout history, countries such as Qi, Lu, Cao, Teng, and Wei had been founded there and achieved great prosperity (Figure 1). The special geopolitical pattern and profound historical culture provided a good basis for the occurrence and development of ICH. Abundant ICH resources had been produced. However, there have been few related types of research to study the spatial distribution and influence factors in Shandong Province. In this study, we applied spatial autocorrelation, cold and hot spot analysis, and standard deviation ellipse analysis to address the distribution pattern of ICH in Shandong Province. ICH grouped area, lacking area, as well as distribution tendency and direction, were identified. Meanwhile, the influence mechanism framework of ICH was constructed based on the man–land relationship. The interaction among influence factors that influenced ICH distribution was addressed through quantitative analysis using the geodetector model. This study could provide a solid basis for the protection and exploitation of ICH.

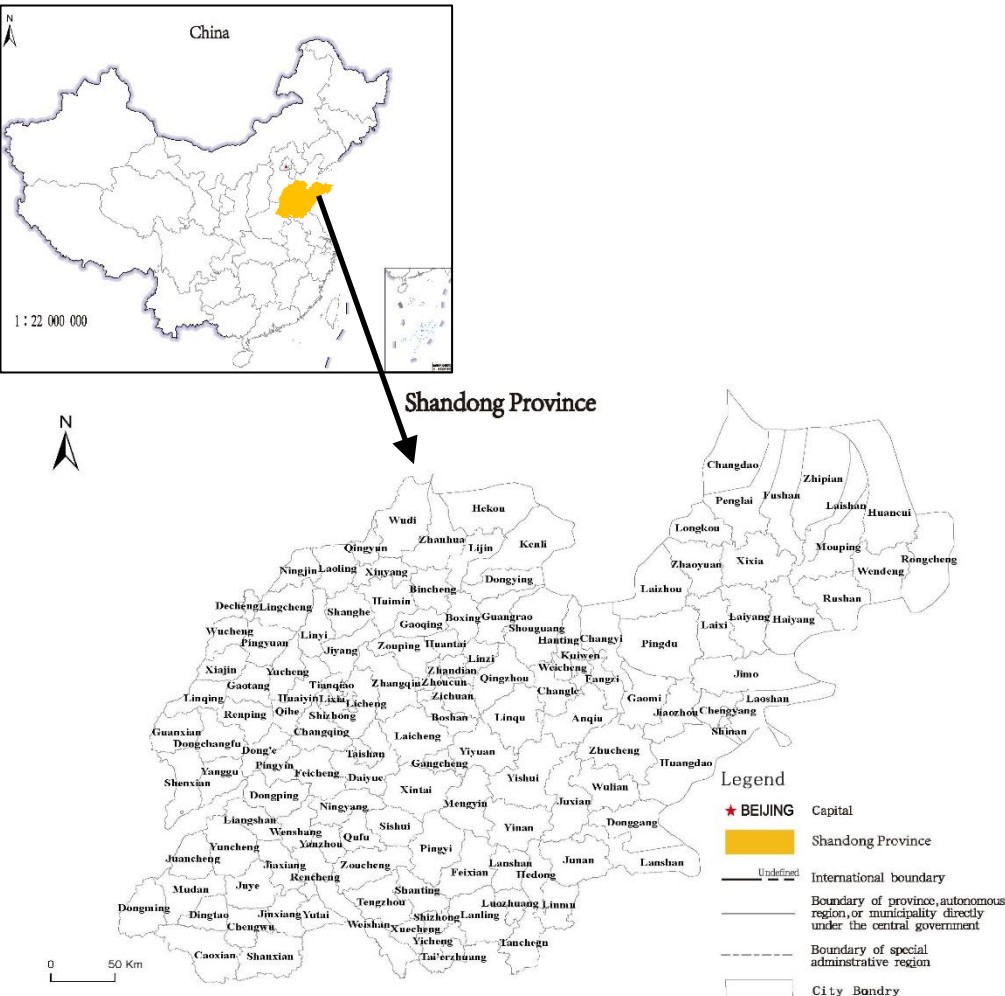

**Figure 1.** The location of the study area in China.

## 2. The Research Framework of ICH Influence Mechanism under the Man–land Relationship Perspective

A man–land relationship is the interaction and feedback process of humans and nature. In the term "man–land", "man" refers to a human that has natural attributes and evolved from nature. It includes the social and economic activity of humans [20]. "Land" means natural environment including landscape, climate, hydrology, land cover, etc. [21]. ICH was generated and developed in the challenge process of humans and nature [22]. Under the man–land relationship perspective, humans and nature have a direct effect on the generation and development of ICH, and their interaction influenced the distribution of ICH.

In the man–land relationship, man is the most active factor [23]. The social and economic activity of humans reflected the influence of human activity on ICH and reflected the restriction of the natural environment to human activity [13]. The social condition of human activity determined the basic personality of humans while the economic condition determined the production manner and efficiency. Both of them restrict the development feature of regional ICH [24].

The natural environment provides basic space for ICH and is the important factor that determines the "point-surface-piece" development pattern and propagation route of ICH [18]. Due to the relative inertia in cultural psychology, ICH is shaped by the natural environment that migrates along with the population in the form of transcendental cultural characteristics. Meanwhile, ICH will change its features along with the social environment [19]. The "Convention for the Safeguarding of the Intangible Cultural Heritage",

initiated by the United Nations Educational, Scientific, and Cultural Organization (UN-ESCO), considered that the natural environment is an important local circumstance that generates ICH [25]. Heterotopia [26] and Topophilia [27] addressed the relevance between invisible culture and natural geographical space from different perspectives. Thus, ICH has spatial properties, and the natural environment is an important carrier of ICH distribution.

Man–land interaction could influence the development and distribution of ICH. Human survival and development are the basis of culture's existence and development, and humans could not survive without the natural environment [19]. The first stage of the man–land relationship is primitive culture. Due to the low productivity of this stage, humans depended too much on nature and were in a natural state of begging nature and perceiving nature to be fantastical. At this stage, there were no words; humans could only attribute success or failure in production and life to the mysterious power in nature, and they worshiped nature [28]. Many gods were created in legends, gradually forming expressions of ICH such as folk literature [19].

The second stage of a man–land relationship is agricultural civilization, which occurred when humans started to use the instrument of labor. The relationship between man and nature in this stage is in a low-level balanced state. A human can partly break the restriction of nature, thus increasing social organizing ability and agricultural productivity. To celebrate such success, traditional music, traditional dance, traditional medicine, and folk customs were generated [18].

In the third stage of the man–land relationship, along with the great geographical discovery, the world becomes an entirety. Great improvement was made in the ability of humans to understand and alter nature. Followed by industry reformation, productivity achieved extreme development. Cross-regional culture communication promoted the improvement and development of ICH [19].

In modern society, ICH is the experience reconstruction of historical memory which reflects the insertion of historical space into real space. The process from memory to experience is the transformation process from real space to inner emotion. It is also the transformation process from historical ICH to the ICH that could be accepted by the human spirit [29]. The inheritance of ICH is one type of spirit value that condensed the experience and memory which could be sublimated to an aesthetic image [30]. Starting from this point of view, the modern ICH is the bridge between humans and the natural environment. The distribution and development of ICH provides emotional support and symbolic metaphors for the regional identity of humans, as well as space and content for the connection between man and land [31]. Based on their cultural identity, humans achieve their identity in society and economy through blood relationships, colleague relationships, and regional relationships, which promote the continuous development of human society and economy [32].

In the relationship between man and land, human activity has a subjective initiative, which can understand, utilize, and transform nature actively. As the material basis and space carrier for human survival, the depth, breadth, and speed of human activity are restricted by the natural environment. Humans must continuously explore the laws of nature to maintain survival. The diversity of the natural environment leads to the formation of different ways of human life and behavior. This difference in human activity superimposed on the natural environment forms the differences in ICH. The existing ICH contains the process of exploring the nature of humans, which is the result of the temporal balance of the man–land challenge.

In summary, the distribution of current ICH is influenced by human activity and natural factors, as well as the interaction of those factors. According to the meaning of the man–land relationship, human activity factors included two main types: social and economic factors. By combing the relevant research on the influencing factors of the spatial distribution of ICH, we selected the population size, transportation, and farmland area as the social factors, and included economic development and urban–rural size as economic

factors. Meanwhile, natural factors that influenced the distribution of ICH were selected from parameters related to climate, topography, forest, and waters (Figure 2).

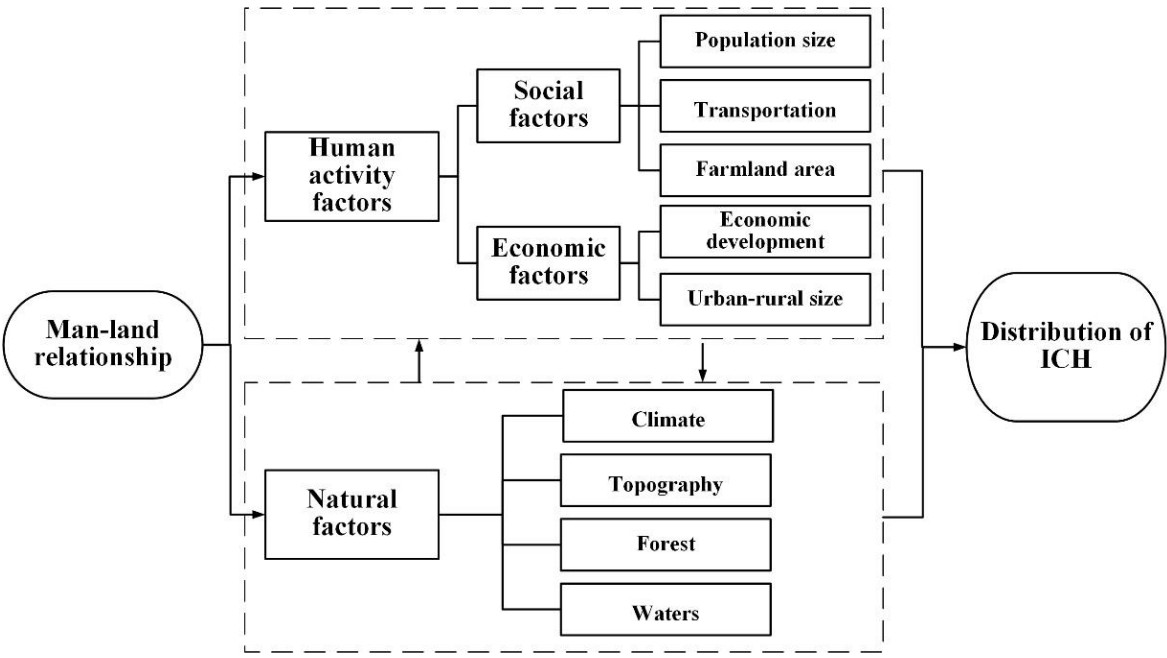

**Figure 2.** Theoretical framework.

*2.1. Human Activity Factors Selection*

2.1.1. Social Factors

(1) Population Size

The distribution of ICH is closely related to population size because people are the carriers of ICH [18]. ICH comes from creation, blood relationships, ancestral authority, and traditional beliefs [19]. The increase in population leads to a grouping that can promote the development of ICH [33]. Thus, population size was selected as one of the social factors (Table 1).

**Table 1.** Influence factors of ICH distribution.

| Category | Primary Index | Secondary Index | Variable Declaration | Unit | Data Source |
|---|---|---|---|---|---|
| Human activity factor | Social factor | Population size | Population size | | "Annals of Statistics 2021" |
| | | Transportation | The number of road and railway | | OSM data https://wiki.openstreetmap.org/wiki/Main_Page (accessed on 16 December 2021) |
| | | Farmland area | Farmland area | hm² | "Annals of Statistics 2021" |
| | Economic factor | Economic development | GDP | $1 \times 10^4$ Yuan | "Annals of Statistics 2021" |
| | | Urban–rural size | Urban–rural construction land area | hm² | "Annals of Statistics 2021" |
| Natural factor | Climate | Air temperature | Average annual temperature | ° | https://www.resdc.cn/ (accessed on 18 December 2021) |
| | Topography | Altitude | Mean DEM | m | https://www.resdc.cn/ (accessed on 18 December 2021) |
| | Forest | Forest area | Forest area | hm² | https://www.resdc.cn/ (accessed on 18 December 2021) |
| | Waters | Water area | Water area | hm² | https://www.resdc.cn/ (accessed on 18 December 2021) |

(2) Transportation

The distribution of ICH was closely related to transportation because the mobility of transportation improves the link between the local area and its surrounding area, which could alter the distribution of regional ICH [18]. Roads and railways are the most important transportation methods; thus, the number of roads and railways were selected as indicators to quantitatively measure transportation (Table 1).

(3) Farmland Area

Farmland area is important in the distribution of ICH because civilization originated from agriculture [34]. The interaction of man and land was reflected by investment and outcome. Farmland is the primary product of man–land interaction [35]. The spatial pattern of farmland determined the spatial structure of the man–land relationship [36]. Therefore, in essence, a man–land relationship is founded based on the development of agriculture and the occurrence of similar culture and management structures. Thus, farmland area was selected as one of the social factors (Table 1).

2.1.2. Economy

(1) Economic Development

Modern ICH was founded based on higher spiritual demands that came after the economic demands had been satisfied [37]. First of all, according to Maslow's hierarchy of needs, the more developed an economy, the higher the spiritual demand. People and government may pay more attention to the protection and development of ICH in the region with a developed economy. Secondly, with economic growth, the urban development pattern will change to multidimensional upgrading. ICH can not only reshape the quality and connotation of cities and strengthen the characteristics of cities, but also promote social inclusion and identity, and stimulate social cohesion and vitality. At the same time, due to the economic benefits of culture, the development of ICH can promote production and consumption, and promote the transformation and upgrading of the urban economy. Therefore, the government is committed to the protection and utilization of ICH in order to achieve the purpose of promoting the multidimensional development of the city. Once more, the developed economy provides the necessary technology and material basis for ICH [10]. Thus, GDP was selected as an indicator to evaluate economic development (Table 1).

(2) Urban–rural Size

City and rural area are the main place to generate ICH [38]. The grouped large-scale urban–rural area that formed in the social and economic development process provides space for ICH generation [38]. The larger the urban–rural size, the easier the foliated distribution pattern of ICH is formed [39]. At present, China is in a new era of urban–rural integration and development. The government is actively promoting cultural exchanges between urban and rural areas. Under the guidance of the principle of city leading township, the government promotes the equalization of the development of urban and rural ICH and other cultural resources in order to achieve the purpose of the high-level coupling state of equal coexistence, benign interaction, and harmonious coexistence of two different ICH forms of city and township [40]. At the same time, the land system as a carrier can drive the flow of ICH between cities and townships and improve the level of urban–rural integration. Thus, urban–rural construction land was selected as one of the economic factors (Table 1).

*2.2. Natural Influence Factors Selection*

(1) Climate

A warm climate is good for human survival and for socioeconomic development, as well as for the succession of ICH [18]. In Chinese history, the period of cultural prosperity was consistent with the period when the climate was warm [41]. Thus, we selected the annual mean temperature as an indicator.

(2) Topography

The climate and water–soil conditions were different at different topographies, which influenced the distribution of ICH. The high altitude led to high water velocity, which may have aggravated the water and soil loss, making it difficult for agricultural production and house construction. Human culture activity may have reduced, which decreased the number of expressions of ICH [42]. Thus, we selected altitude as an indicator (Table 1).

(3) Forest

Forest could provide material guarantee for ICH. Areas with high forest cover usually contain abundant biological resources which could provide sufficient food sources [43]. Meanwhile, forests could provide raw materials, promoting the human population in the same region by speeding up production [44]. Therefore, forests helped to increase population and succeed in ICH [45]. Thus, we selected forest area as an indicator (Table 1).

(4) Water

Water is an important factor that influences the distribution of ICH, as well as an important material basis for human production and life [46]. Regions with more abundant water resources usually bore more elements of ICH due to active cultural activity in such areas [19]. Ancient civilizations more or less started and developed along riversides [19]. Therefore, we selected water area as an indicator (Table 1).

## 3. Methods

In this study, the administrative region (county and city) was set as a research unit. Data including the total amount and the number of each type of ICH were obtained from national- and provincial-level records. The distribution pattern of ICH was analyzed by spatial autocorrelation, hot spot clustering, and standard deviation ellipse analysis. The influence mechanism of man–land relationship indicators was analyzed through a geodetector model.

### 3.1. Distribution Pattern Analysis

3.1.1. Spatial Autocorrelation

Global Moran's I index was used to test the correlation between two adjacent units in the study area [47]. The calculation equation is as follows:

$$I = \frac{\sum_{i=1}^{n} \sum_{j=1}^{n} W_{ij} \left( Y_i - \overline{Y} \right) \left( Y_j - \overline{Y} \right)}{S^2 \sum_{i=1}^{n} \sum_{j=1}^{n} W_{ij}} \tag{1}$$

$$Z(I) = \frac{I - E(I)}{\sqrt{Var(I)}} \tag{2}$$

$I$ means Global Moran's I index; $n$ indicates the number of sample unit; $Y_i$ and $Y_j$ mean the number of expressions of ICH in units $i$ and $j$; $W_{ij}$ means the spatial weight matrix (when two units are spatially adjacent, the value is 1, otherwise 0); $S^2$ is the variance of the ICH number; $\overline{Y}$ is the average number of expressions of ICH. The value range of I is −1 to 1. $I > 0$ indicates positive spatial correlation; $I < 0$ indicates negative spatial correlation; $I = 0$ indicates no spatial correlation. $Z(I)$ is the Z test result of Moran's $I$; $E(I)$ is expected value; $Var(I)$ is the coefficient of variation.

3.1.2. Cold and Hot Spot Analysis

Getis-Ord $G_i^*$ was used to recognize the significantly high (hot spot) and low (cold spot) values in the ICH distribution in Shandong Province. $G_i^*(d)$ the index could be used to recognize the local spatial autocorrelation feature [47]. The statistical meaning of $G_i^*(d)$ was measured by the normalized $Z\left(G_i^*\right)$ value.

$$G_i^*(d) = \frac{\sum_{j=1}^{n} w_{ij}(d) X_j}{\sum_{j=1}^{n} X_j} \tag{3}$$

$$Z(G_i^*) = \frac{G_i^* - E(G_i^*)}{\sqrt{Var(G_i^*)}} \tag{4}$$

In the above equation, $w_{ij}$ is the spatial weight matrix, $X_j$ is the number of expressions of ICH in the study unit j, and $E(G_i^*)$ and $Var(G_i^*)$ indicate the expected value and coefficient of variance of $G_i^*$. $Z(G_i^*) > 0$ means the number of expressions of ICH in the ith region was higher than the average value; $Z(G_i^*) < 0$ means the number of expressions of ICH in the ith region was lower than the average value.

3.1.3. Standard Deviation Ellipse Analysis

Standard Deviational Ellipse (SDE) analysis was applied to address the distribution direction of ICH in Shandong Province. The core of SDE means the barycentric coordinate of elements distribution. The azimuth angle means the horizontal angle of element distribution.

(1) Core

$$\overline{X} = \frac{\sum_{i=1}^{n} w_i x_i}{\sum_{i=1}^{n} w_i}, \overline{Y} = \frac{\sum_{i=1}^{n} w_i y_i}{\sum_{i=1}^{n} w_i} \tag{5}$$

(2) Azimuth angle:

$$\tan \theta = \frac{\left(\sum_{i=1}^{n} w_i^2 \tilde{x}_i^2 - \sum_{i=1}^{n} w_i^2 \tilde{y}_i^2\right) + \sqrt{\left(\sum_{i=1}^{n} w_i^2 \tilde{x}_i^2 - \sum_{i=1}^{n} w_i^2 \tilde{y}_i^2\right)^2 + 4 \sum_{i=1}^{n} w_i^2 \tilde{x}_i \tilde{y}_i}}{\sum_{i=1}^{n} 2 w_i^2 \tilde{x}_i \tilde{y}_i} \tag{6}$$

In the above equation, $n$ is the number of sample units; $x_i$ and $y_i$ are the longitude and latitude geographic coordination of the core in the $i$th sample unit; $w_i$ is the number of expressions of ICH; $\overline{X}$ and $\overline{Y}$ are the weighted average core coordination; $\theta$ is the azimuth angle of SDE, indicating the distribution direction of ICH.

*3.2. Geo Detector*

The influence mechanism of each factor was analyzed by a geodetector.

$$q_j = 1 - \frac{\sum_{h=1}^{L} N_{jh} \eth_{jh}^2}{N_j \eth_j^2} \tag{7}$$

In the above equation, $h = 1, \dots L$ are the region order of factor j; $N_{jh}$ and $N_j$ indicate the number of units in region $h$ and whole region for factor $j$; $\eth_j^2$, $\eth_{jh}^2$ mean total variance in the whole region and total variance in region $h$ for factor $j$; $q_j$ is the influence of factor $j$ on the spatial heterogeneity of the number of expressions of ICH; $q_j \in [0,1]$ and large $q_j$ values mean strong influence. The interaction between factors could be detected by comparing the effect of multiple dependent variables on an independent variable and a single dependent variable on the same independent variable [48]. The P Value Test reflects the influence of factor $j$ on the number of expressions of ICH. There are five types of effects between two independent variables:

(1) $q(x_1 \cap x_2) < min(q(x_1), q(x_2))$ indicate that the nonlinear relation for both factors decreased after $x_1$ and $x_2$ interacted;

(2) $q(x_1 \cap x_2) > max(q(x_1), q(x_2))$ indicate that the influence of both factors increased after $x_1$ and $x_2$ interacted;

(3) $min(q(x_1), q(x_2)) < q(x_1 \cap x_2) < max(q(x_1), q(x_2))$ indicate that the nonlinear relation for a single factor decreased after $x_1$ and $x_2$ interacted;

(4) $q(x_1 \cap x_2) > q(x_1) + q(x_2)$ indicate that the nonlinear relation for both factors increased after $x_1$ and $x_2$ interacted;

(5) $q(x_1 \cap x_2) = q(x_1) + q(x_2)$ indicate the independent relation for both factors $x_1$ and $x_2$.

*3.3. Data Source and Treatment*

National ICH data were obtained in the years 2006, 2008, 2011, 2014, and 2021. Provincial-level data were obtained in the years 2006, 2009, 2013, 2016, and 2021. According to the Chinese national ICH list, the ICH was divided into 10 types: folk literature, traditional music, traditional dance, traditional drama, traditional quyi, traditional sports, recreation and acrobatics, traditional art, traditional skills, traditional medicine, and folk custom. Influence factor data such as population size, economic development, farmland area, urban–rural construction land area, and forest area were obtained from government records. Altitude data were obtained from the Data Center for Resources and Environmental Sciences of the Chinese Academy of Sciences. The average annual temperature was obtained from the Institute of Geographic Sciences and Natural Resources Research. The number of roads and railways was obtained from OSM data.

**4. Results**

*4.1. Analysis of the Number of Expressions of ICH*

There is a full range of ICH types with different numbers of each type in Shandong Province. The total number of expressions of ICH is 1092, of which 189 are national, and 903 are provincial (Table 2). In the national ICH, the main types are traditional drama, folk literature, traditional art, and traditional skills. In the provincial-level ICH, the main types are traditional skills, traditional art, traditional sports, recreation and acrobatics, and traditional dance.

**Table 2.** The type and number of expressions of ICH.

| Type of ICH | National Level | Provincial Level | Total |
|---|---|---|---|
| Traditional skill | 21 | 313 | 334 |
| Traditional art | 28 | 116 | 144 |
| Traditional sports recreation and acrobatics | 15 | 80 | 95 |
| Traditional dance | 12 | 69 | 81 |
| Traditional drama | 33 | 45 | 78 |
| Traditional medicine | 4 | 61 | 65 |
| Traditional music | 19 | 47 | 66 |
| Fold literature | 30 | 64 | 94 |
| Folk custom | 14 | 61 | 75 |
| Traditional quyi | 13 | 47 | 60 |
| Total | 189 | 903 | 1092 |

Overall, ICH in Shandong Province mainly consists of traditional skills, traditional art, traditional sports, recreation and acrobatics, and folk literature, accounting for 30.59%, 13.19%, 8.70%, and 8.61% of the total amount, respectively. The percentage of traditional dance, traditional drama, traditional medicine, traditional music, folk custom, and traditional quyi were all below 8.00%, of which traditional medicine and folk custom account for 5.59% and 5.49%, respectively.

*4.2. Analysis of the Spatial Distribution of ICH*

4.2.1. Spatial Autocorrelation

The results of spatial autocorrelation showed that: the Moran's I index value was 0.16, therefore larger than 0; the z-value was 3.03 ($p < 0.01$). It indicates the possibility that ICH in Shandong Province distributed in the grouped pattern was less than 1%. The distribution of ICH showed a clear positive spatial correlation. Both the region with large and small numbers of expressions of ICH formed the pattern of the spatial group (Figure 3).

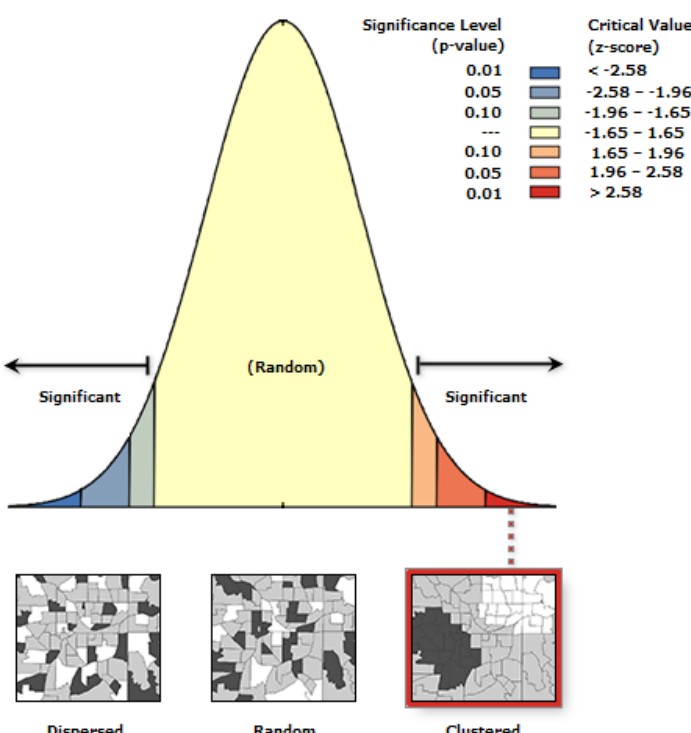

**Figure 3.** The analysis results of spatial autocorrelation of ICH in Shandong Province.

### 4.2.2. Cold and Hot Spot Analysis

Cold and hot spot analysis was applied to investigate the group feature of ICH in Shandong Province. Figure 4 showed that the distribution of the ICH pattern was agglomerated with three hot spots and four cold spots. The trinuclear structure was shown in the group area of ICH. The first one is the middle core group area, which was concentrated in Jinan City and Zibo City. A total of 119 expressions of ICH were found in this area, accounting for 10.90% of the total number of expressions of ICH in Shandong Province. The distribution showed a step-shaped pattern. The second one is the southwest group area where Heze City is the center. The total number of expressions of ICH in this area was 79, accounting for 7.23% of the total number. The third one is the northeast group area where Yantai City is the center. The total number of expressions of ICH is 50, accounting for 4.58% of the total number. The four cold spots are Pingyuan Country, Linyi Country, Hekou District, and Laixi City with ICH numbers of 4, 2, 1, 1, and 2, respectively. Those areas are ICH lacking areas (Figure 4).

### 4.2.3. Standard Deviation Ellipse Analysis

The results of SDE showed that: the distribution center of ICH in Shandong Province was located in the middle core group area, which almost covered the whole middle part of Shandong Province, including 90 districts (country and city). The azimuth angle was 61.09°, showing the distribution direction from southwest to northeast. The long axis of SDE ran along the Yellow River (Figure 5).

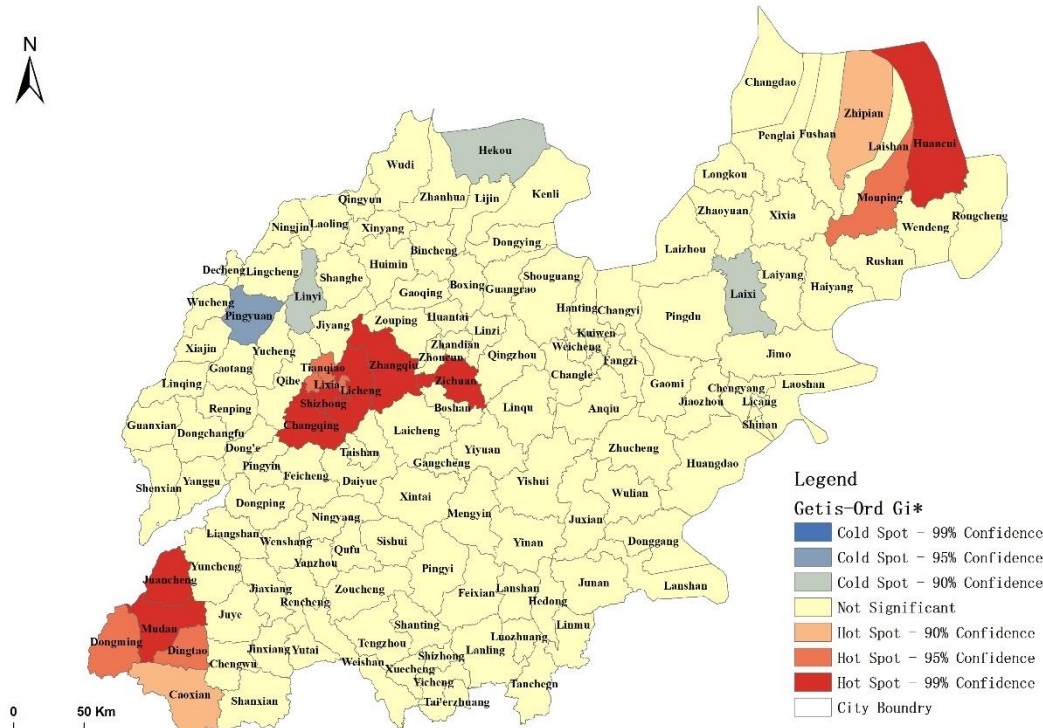

**Figure 4.** The analysis results of cold and hot spot analysis of ICH in Shandong Province.

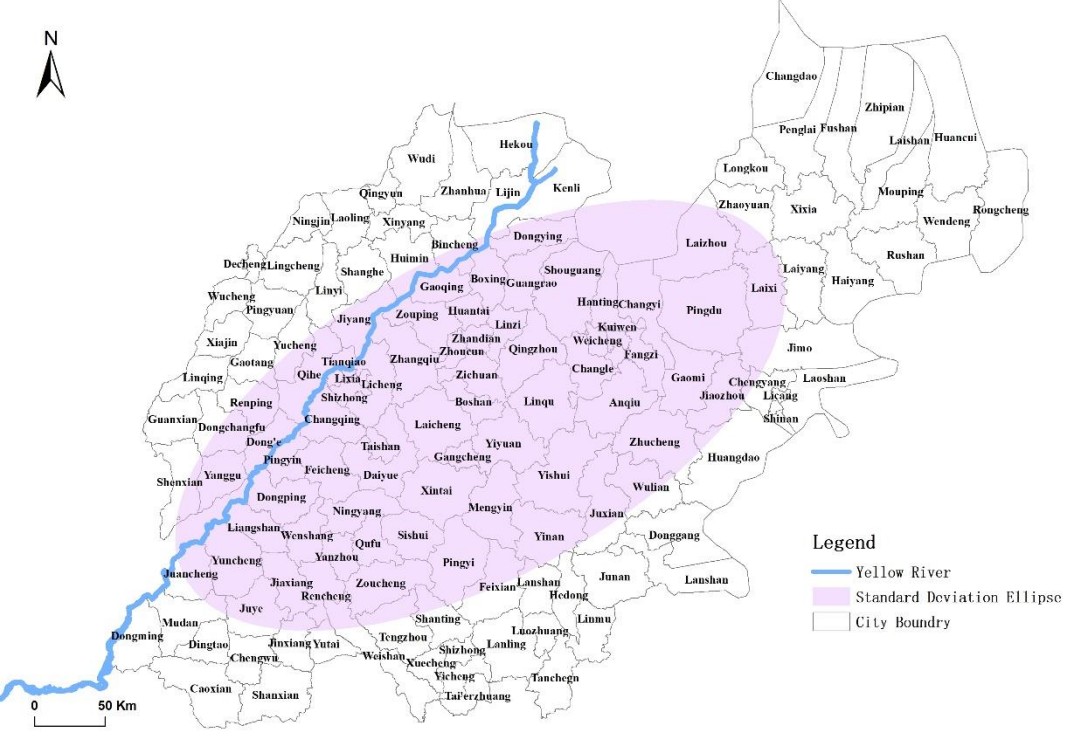

**Figure 5.** Spatial distribution map of ICH in Shandong Province.

In summary, the spatial difference is clear in the ICH distribution in Shandong Province, showing the pattern of multiple group areas and lacking areas. The distribution of ICH followed the direction from southwest to northeast.

*4.3. Influence Factors Analysis*

In this study, the number of expressions of ICH was defined as the dependent variable, while population size, the number of roads and railways, farmland area, GDP, urban–rural construction land area, air temperature, altitude, forest area, and water area were defined as independent variables. The geodetector was used to measure the influence of different indicators on the distribution of ICH. The results showed that: different indicators present diverse influences on the distribution of ICH in distinct dimensions.

4.3.1. Single Factor Detection

There are great differences in cultural and physical geographical conditions in Shandong Province. The spatial distribution of ICH was influenced by multiple factors and the interaction among them. The q-value of each factor, ordered from high to low, is as follows: population size, water area, urban–rural construction land area, air temperature, the number of roads and railways, forest area, GDP, farmland area, and altitude. Population size, water area, urban–rural construction land area, and air temperature are the key driving factors for ICH distribution, while the influences of the number of roads and railways, forest area, GDP, and altitude are not strong. The q-value of each human activity factor, ordered from high to low, are as follows: population size, urban–rural construction land area, the number of roads and railways, GDP, and farmland area ($p < 0.001$). The q-value of each natural factor that is ordered from high to low is as follows: water area, air temperature, forest area, and altitude (Table 3).

**Table 3.** Results of single factor detection.

| Indicator | q | p |
|---|---|---|
| Population size | 0.26 | 0.00 |
| Transportation | 0.11 | 0.00 |
| Farmland area | 0.10 | 0.00 |
| Economic development | 0.10 | 0.00 |
| Urban–rural area | 0.19 | 0.00 |
| Air temperature | 0.15 | 0.00 |
| Altitude | 0.09 | 0.00 |
| Forest area | 0.10 | 0.00 |
| Water area | 0.19 | 0.00 |

4.3.2. Detection of the Interaction between Factors

The results of interaction between factors showed that the influence of human activity factors and natural factors on ICH distribution are not independent, but two factors enhance the way. The interactions between man–land factors all show a stronger influence than a single factor on the distribution of ICH. The interactions between man–land factors are more capable of explaining the difference in ICH distribution. However, the influence of interactions between different factors varied.

The interaction between population size and forest area showed the strongest influence on the distribution of ICH (q = 0.67, explanatory power is around 67%). Population and forest area are the fundamental bases of ICH generation and development, which were important in supporting ICH to succeed. The paired factors that have explanatory power higher than 50% were population size and water area, economic development and forest area, economic development and water area, transportation and water area, transportation and forest area, and urban–rural size and water area (Table 4).

**Table 4.** The influence of interactions between factors and ICH Distribution.

| q＼Indicator ／ Indicator | Popula-tion Size | Transporta-tion | Farm-land Area | Economic Develop-ment | Urban–Rural Size |
|---|---|---|---|---|---|
| Air temperature | 0.42 | 0.48 | 0.33 | 0.38 | 0.48 |
| Altitude | 0.36 | 0.41 | 0.27 | 0.32 | 0.43 |
| Forest area | 0.67 | 0.51 | 0.39 | 0.52 | 0.47 |
| Water area | 0.53 | 0.52 | 0.43 | 0.57 | 0.57 |

## 5. Discussion

### 5.1. The Number of Expressions of ICH

With the help of quantitative analysis, it was found that there is a full range of ICH types with different numbers of each type in Shandong Province. The ICH in Shandong Province mainly consisted of traditional skills, traditional art, traditional sports, recreation and acrobatics, and folk literature. Shandong is one of the birth lands of Chinese ancient culture. A large number of philosophers, politicians, and scientists such as Kong Zi and Meng Zi contributed their talents to the generation and development of ICH through elements such as folk literature, traditional art, traditional music, traditional drama, and traditional medicine. Meanwhile, the iron smelting industry developed in the early stage of Shandong Province, which can track back to the first year of the Spring and Autumn period (more than 2000 years ago), and Shandong Province was the major iron smelting center in the Northern Song Dynasty. The textile and handicraft industry of Shandong Province are world-famous. During the Warring States period, Shandong Province was known as "the world with clothes and shoes". Shandong Province is also one of the main sources of the "Silk Road". Linzi District, Dingtao District, and Kangfu City (now Jining City) were the three major textile centers in the Han Dynasty, and Jinghuaxu and Xianwen damask were famous textiles in the Tang Dynasty. All of those provided abundant material for the development of traditional skills.

### 5.2. The Spatial Distribution of ICH

Through the spatial pattern analysis, this study found that the distribution of ICH in Shandong Province has a great spatial heterogenicity. The ICH agglomeration area in Shandong Province presents three gathering hotspots, which is consistent with the research results on the spatial distribution characteristics for multicore spatial structures in other regions [3,4]. Compared with the previous studies, the present study distinguished that effectively, there are four cold spots of ICH distribution in Shandong Province. According to the result, some measures of intangible cultural heritage development could be conducted in these areas.

### 5.3. Influence Factors of ICH

5.3.1. Single Factor Detection

(1) Human Activity Factors

The results indicate that human activity factors greatly influenced the distribution of ICH. This result proves that ICH is the product of human activity, and humans decide the production and distribution of ICH.

Concretely, the explanatory power of population size is the highest (q = 0.26), which indicates population size is the most important factor that influenced the distribution of ICH (Table 3). Human activity is the carrier of ICH; thus, regions that have a big population usually have more demand for ICH, where it is easier to form an ICH group area. The

total population in the four cold-spot areas is only 1.99 million, accounting for 1.79% of the population in Shandong Province. The small population size is the main reason that those areas become cold-spot areas of ICH.

The explanatory power of urban–rural construction land area is relatively high (q = 0.19), which indicates that urban–rural construction land area is an important factor that influences the distribution of ICH (Table 3). Urban–rural construction land is an important carrier of social and economic activity, providing space for ICH activity. For instance, the urban–rural construction land area in Licheng District, Mudan District, and Cao Country are all larger than 30,000 hectares, which provide plenty of space for ICH activity.

The q-value of the number of roads and railways is 0.11, which indicates that the ICH grouped area is mainly located in the region with developed transportation (Table 3). The determination of ICH considered the cost of protection. The region with developed transportation could promote the combination of ICH and tourism, which could reduce the cost of ICH protection. Thus, the distribution of ICH was significantly correlated with transportation. For instance, in the southwest grouping area, the center region is Heze City, which has developed roads and railways. The developed transportation promotes the formation of hot-spot areas regarding ICH distribution.

The q-value of GDP is 0.11, which indicates that economic development has a significant influence on the distribution of ICH. Economic development promotes competition between cultures, which made the culture develop more greatly in the vertical direction and in a more diverse way in the horizontal direction. The hot-spot areas are all located in an economically developed region that can support the ICH development.

The q-value of farmland area is 0.10, which indicates that farmland area has a significant influence on the distribution of ICH (Table 3). Shandong Province has developed an agricultural industry since ancient times, which provides a solid basis for the generation of ICH. In the southwest group area (Heze City), the farmland area is at the top level of the province. This greatly promotes the generation and development of ICH.

(2) Natural Factors

The results of natural factors show that the most important natural factor is the water area, with a q-value of 0.19 (Table 3). Water areas have the function of shipping, irrigation, flood control, and cultivation, which provided great convenience to human life in ancient times. Shandong Province has many rivers and lakes, especially the Yellow River and the Grand Canal, which flow throughout the whole province. In addition to the rivers and lakes, the peninsula is surrounded by the ocean on three sides. Those waters greatly promote the generation and development of ICH. In this study, the two hot spots, the middle core group area and the southwest group area, are all located around the Yellow River. The other hot spot, the northeast group area, is surrounded by the Bo Ocean. Those results prove that water area is important to the generation and development of ICH.

The q-value of air temperature is 0.14, which indicates that air temperature has a significant influence on the distribution of ICH (Table 3). The climate in Shandong Province is a temperate monsoon climate with a short spring and autumn, and long summer and winter. The average annual temperature is 11–14°C, which is good for human production and life. This helped the development of ICH. Meanwhile, the air temperature difference is clear, and the east to the west line is larger than the south to the north line. The hot-spot area has less-extreme weather, which made the number of expressions of ICH higher there than in other regions.

The q-value of forest area is 0.10, which indicates that forest area has a significant influence on the distribution of ICH (Table 3). The forest area in Shandong Province in 2020 was 2,665,100 hm$^2$. The forest is mainly distributed in the middle and east parts, which is consistent with the hot-spot area, proving that the forest could promote the development and protection of ICH.

The q-value of altitude is 0.08, which indicates that altitude has a relatively weak influence on the distribution of ICH (Table 3). The landscape in Shandong Province mainly consists of plain and upland, which account for 63% and 34%, respectively. The plain and

upland show a mosaic distribution, which make a diverse topography. For instance, the southwest group area is located in the plain, where the cultural communication and spread is convenient, promoting the development of ICH.

### 5.3.2. Detection of the Interaction between Factors

The interaction results indicate that a region with a high forest area, water area, and urban–rural construction land area could hold higher population and economic activity, which can promote the generation and development of ICH. The human activity factors that were superimposed on natural elements showed a stronger influence on the distribution of ICH, which indirectly proved that the distribution is the result of man–land interaction.

The southwest group area in Shandong Province does not have a high degree of economic development or high forest area. However, the interactions between population size, transportation, urban–rural size, water area, and farmland area made a good condition for the generation and development of ICH. Nevertheless, in the middle core group area, the interaction between population size, economic development, transportation, water area, and forest area broke the restriction of advent landscape factors. Meanwhile, although the cold spot area in Pingyuan Country, Linyi Country, and Hekou District is located in good topography with plenty of water area and developed transportation, the hot spot could not be formed even after interactions due to the restriction of the low economic development, small urban–rural size, and low forest area.

On the whole, population size, transportation, farm-land area, economic development, urban–rural size, air temperature, altitude, forest area, and water area showed significant influence on the distribution of ICH. This was in line with the previous studies of Li et al. (2017) [17] and Anderson et al. (2002) [18]. The most interesting finding is that the interactions between man–land factors all show a stronger influence than a single factor on the distribution of ICH. This result could provide a new clue for the protection and development of ICH. The research framework of the ICH influence mechanism constructed based on the man–land relationship perspective is a big improvement to current studies.

### 5.4. Suggestions for Policy Making

The number, type, and spatial distribution of expressions of ICH in Shandong Province are different. Considering the number of expressions of ICH, more attention should be paid to traditional dance, traditional drama, traditional medicine, traditional music, folk custom, and traditional quyi by the government. Considering the spatial distribution of expressions of ICH, the government should set up an evaluation and protection system to guarantee the sustainable development of ICH in the hot-spot area. In the provincial-level and national-level ICH evaluation and exploration, the government should assign more chances to the cold-spot area. The current ICH is mainly distributed along the Yellow River and the Grand Canal. Therefore, in the construction of the Yellow River culture band and Grand Canal culture band, the government should consider protection and exploration in a broader area. The distribution difference of ICH in Shandong Province is quite clear. In the exploration of ICH, the government should enhance the cooperation of ICH between different regions and finally fulfill the "point-surface-piece" combined development in different regions. The distribution of ICH in Shandong Province is the result of the interactions between man–land factors. Therefore, the government could not only focus on the protection and exploration of ICH itself. The protection of the man–land relationship must be considered as well. The local government in the cold-spot region should enhance economic power, attract population, protect farmland, improve transportation, increase forest area, guarantee water resources, and comprehensively improve the regional conditions for ICH. Finally, the succession of ICH should also be considered. ICH succession education could help to enhance the protection awareness of the public.

In this study, the migration of development focus in the ICH succession was not considered. In a future study, time factors should be involved in the analysis to obtain a more comprehensive understanding. Furthermore, restricted by evaluation standards,

world level and city level ICH were not included in this study. In the future, we may try to normalize the evaluation standard to include more expressions of ICH and extend the study to the national scale.

## 6. Conclusions

(1) The ICH in Shandong Province covered most ICH types, with differences in the various types of expressions of ICH. The total number of expressions of ICH is 1092, of which 189 are national, and 903 are provincial. ICH in Shandong Province mainly consists of traditional skills, traditional art, traditional sports, recreation and acrobatics, and folk literature, accounting for 30.59%, 13.19%, 8.70%, and 8.61% of the total number, respectively. The percentage of traditional dance, traditional drama, traditional medicine, traditional music, folk custom, and traditional quyi were all below 8.00%, of which traditional medicine and folk custom account for 5.59% and 5.49%, respectively.

(2) The spatial distribution of ICH in Shandong Province shows clear group features. Multiple concentration areas and deficient areas were presented which followed the direction from southwest to northeast, showing a clear positive spatial correlation. The distribution of the ICH pattern was agglomerated with three hot spots and four cold spots. The trinuclear structure was shown in the group area of ICH. The distribution center of ICH in Shandong Province was located in the middle core group area, which almost covered the whole middle part the Shandong Province, including 90 districts (country and city). The azimuth angle was 61.09°, showing a distribution direction from southwest to northeast. The long axis of SDE ran along the Yellow River.

(3) The results of the influence factor analysis showed that: The interactions between man–land factors all show a stronger influence than a single factor on the distribution of ICH. In the single factor analysis, the q-value of each human activity factor, ordered from high to low, are as follows: population size, urban–rural construction land area, the number of roads and railways, GDP, and farmland area ($p < 0.001$). The q-value of each natural factor, is ordered from high to low, is as follows: water area, air temperature, forest area, and altitude. The results of the interaction between factors showed that the influence of human activity factors and natural factors on ICH distribution are not independent, but two factors enhance the way. The paired factors that have an explanatory power higher than 50% were population size and forest area, population size and water area, economic development and forest area, economic development and water area, transportation and water area, transportation and forest area, and urban–rural size and water area.

**Author Contributions:** Conceptualization, L.M.; methodology, L.M.; software, L.M.; validation, L.M; formal analysis, L.M.; investigation, L.M.; resources, L.M.; data curation, L.M.; writing—original draft preparation, L.M.; writing—review and editing, L.M., C.Z., J.P., and B.W.; visualization, L.M.; supervision, L.M. and W.S.; project administration, L.M.; funding acquisition, L.M. All authors have read and agreed to the published version of the manuscript.

**Funding:** This research was funded by Key Program of Pedagogy of National Social Science Foundation of China (Grant NO.AFA 190018) and Philosophy and Social Science Project of Jinan, China (Grant NO. JNSK21C13).

**Data Availability Statement:** Not applicable.

**Conflicts of Interest:** The authors declare no conflict of interest.

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
