# Peer review of "Study on the Influence Mechanism of Intangible Cultural Heritage Distribution from Man–Land Relationship Perspective: A Case Study in Shandong Province"

_land, doi:10.3390/land11081225_

Round 1

Reviewer 1 Report

REVIEWER'S DETAILED SUGGESTIONS

ABSTRACT:

The abstract MUST be structured as follows; and do not need to be numbered:

1. Purpose (mandatory)

2. Design/methodology/approach (mandatory).

3. Findings and Originality/value (mandatory)

4. Research limitations/implications (if applicable)

5. Practical implications (if applicable)

6. Academic implications/Further research (mandatory)

KEYWORDS:

1. Select the most relevant words / 2-word phrases to the research purpose and finding

INTRODUCTION:

Introduction MUST discuss the following areas:

1. Background (ontology / the existing facts relating to the case being examined, as so liable to be research academically)

2. Research Objective (MANDATORY); and MUST be synchronised with the research method, theoretical framework, discussion/finding and conclusion.

3. The reasons why the research questions liable to be examined

4. Novelty or existing gaps

5. Expected benefits and contribution of the study

LITERATURE REVIEW:

This section MUST discuss the following areas:

1. Previous studies (mandatory)

2. Concept and / or major theory used in the article especially the concept of "intangible cultural heritage distribution" and “Man-land relationship” (mandatory)

RESEARCH METHOD:

This section MUST discuss the following areas (but not limited):

1. Selected location / case (Why they are selected case significant?).

2. What is the sampling procedures selecting the data? Use appropriate references!

3. Any ethics clearance has been obtained for this study?

4. Any pilot study has been conducted before the current research for publication?

5. How did you collect the quantitative data? Why such data collection method is appropriate/best for this research?

6. How did you analysed the collected data?

7. How did you discuss the analysed quantitative data in the discussion/finding section?

FINDING / RESULT

This section MUST discuss the following areas:

1. Finding based on the data gathered and answering all research questions/purposes

2. Write specifically based on the research questions/purposes to ease the readers understand the findings.

DISCUSSION

This section MUST discuss the following areas:

- Reflections of the "current study" compared with the "previous studies" to show the "NOVELTY" of current research.

- Discuss how the research finding contributed to science developments of the existing studies

CONCLUSION

This section MUST discuss the following areas:

1. Researcher's view why the case is interesting to be investigated

2. Write conclusions based on the research questions

3. Research Implications and limitations

4. Suggestion for further scientific research related to this finding

ENGLISH ISSUE

- Sufficient

Author Response

请参阅附件。

Reviewer 2 Report

Intervention is necessary at the level of graphics - the font size is very small and is not perceptible. Inside the map and in the legend - figure 1, figure 4 and figure 5.

In the structure of the article, at 5.2 The spatial distribution of ICH, text appears in Chinese - explain its presence, if applicable, or it must be removed!

Author Response

请参阅附件。

Reviewer 3 Report

i) Suggest to include intro, problem statement, objective, method in abstract

ii) More references on research which similar to your topic of interest 

iii) discuss more on theoretical frameworks.

Author Response

请参阅附件。

Reviewer 4 Report

The study applies global spatial autocorrelation, hotspot analysis, and standard deviation ellipse analysis to address the distribution pattern of Intangible Cultural Heritage in Shandong province, by identifying ICH grouped area, lacking area, as well as distribution tendency and direction. The paper fills the gap related to quantitative analysis and try to understand the influence mechanism framework of ICH  based on the man-land relationship.

The methodology adequately addresses the quantitative analysis using  the geo-detector model.

Nevertheless, the claim of providing a solid basis for the protection and exploitation of ICH by supporting cultural tourism route design needs to be better supported and explained. There is a lack in discussing the interpretation of the research results in terms of applications and policy design.

Man-land relationship needs to be interpreted by including the urban pattern and the rural-urban transition in order to be more consistent with the aims.

Author Response

请参阅附件。
